# Progress to Improve Oral Bioavailability and Beneficial Effects of Resveratrol

**DOI:** 10.3390/ijms20061381

**Published:** 2019-03-19

**Authors:** Adele Chimento, Francesca De Amicis, Rosa Sirianni, Maria Stefania Sinicropi, Francesco Puoci, Ivan Casaburi, Carmela Saturnino, Vincenzo Pezzi

**Affiliations:** 1Department of Pharmacy and Health and Nutritional Sciences University of Calabria, Via Pietro Bucci, Arcavacata di Rende, 87036 Cosenza, Italy; francesca.deamicis@unical.it (F.D.A.); rosa.sirianni@unical.it (R.S.); s.sinicropi@unical.it (M.S.S.); francesco.puoci@unical.it (F.P.); ivan.casaburi@unical.it (I.C.); 2Department of Science, University of Basilicata, Viale dell’Ateneo Lucano 10, 85100 Potenza, Italy; carmela.saturnino@unibas.it

**Keywords:** resveratrol, resveratrol bioavailability, resveratrol delivery systems, resveratrol derivatives

## Abstract

Resveratrol (3,5,4′-trihydroxystilbene; RSV) is a natural nonflavonoid polyphenol present in many species of plants, particularly in grapes, blueberries, and peanuts. Several in vitro and in vivo studies have shown that in addition to antioxidant, anti-inflammatory, cardioprotective and neuroprotective actions, it exhibits antitumor properties. In mammalian models, RSV is extensively metabolized and rapidly eliminated and therefore it shows a poor bioavailability, in spite it of its lipophilic nature. During the past decade, in order to improve RSV low aqueous solubility, absorption, membrane transport, and its poor bioavailability, various methodological approaches and different synthetic derivatives have been developed. In this review, we will describe the strategies used to improve pharmacokinetic characteristics and then beneficial effects of RSV. These methodological approaches include RSV nanoencapsulation in lipid nanocarriers or liposomes, nanoemulsions, micelles, insertion into polymeric particles, solid dispersions, and nanocrystals. Moreover, the biological results obtained on several synthetic derivatives containing different substituents, such as methoxylic, hydroxylic groups, or halogens on the RSV aromatic rings, will be described. Results reported in the literature are encouraging but require additional in vivo studies, to support clinical applications.

## 1. Health Beneficial Effects of Resveratrol (RSV)

Natural products have recently aroused interest within the scientific community for their beneficial effects on several diseases. Preclinical, clinical, and epidemiological studies have shown that consumption of polyphenols contained in cereals, legumes, vegetables, and fruit at high levels, prevents various diseases, including cancer. The most promising candidate is resveratrol (RSV), a natural nonflavonoid polyphenol found in numerous plant species, in particular in grapes, blueberries, and peanuts [1]. RSV or 3,5,4′-trihydroxystilbene consists of two aromatic rings that are connected through a methylenic bridge (Figure 1). It exists as two geometrical isomers—the *trans*-RSV form (Figure 1B) and the *cis*-form—the first having the greater stability and biological activity and the second being less active (Figure 1A). The last form arises from isomerization of the *trans*-form following the breakdown of the RSV molecule due to the action of UV light during the fermentation of grape skins or under high pH conditions [2].

The increased interest for this compound is based on epidemiological studies showing an inverse relationship between moderate consumption of wine and cardiovascular diseases (the so-called “French paradox”) [3] and on in vitro and in vivo studies demonstrating RSV beneficial effects on human health [4]. Several reports evidenced RSV effectiveness on various diseases such as diabetes mellitus, metabolic syndrome, obesity, inflammation, cardiovascular, neurodegenerative, and age-related diseases, as well as cancer [5]. Although the mechanisms by which RSV exerts its beneficial effects have not yet been fully elucidated, it has been reported that it displays antioxidant, anti-inflammatory, cardioprotective, neuroprotective, and antitumor actions (see Table 1 for summary data). 

It is known that polyphenols exert antioxidant effects related to the presence of hydroxylic groups which participate in mechanisms aimed to decrease reactive oxygen species (ROS) and free radicals and to increase endogenous antioxidants biosynthesis [6]. RSV antioxidant properties have been attributed to its capability to reduce copper-catalyzed oxidation [7] and inhibit lipid peroxidation of low density lipoproteins (LDL) [8] and cellular membranes [9]. Other studies demonstrated that RSV decreases intracellular concentration of apolipoprotein B (ApoB), cholesterol esters, and triglyceride secretion rate, thus protecting against atherosclerosis [10]. RSV anti-inflammatory effects are mainly due to the inhibition of cyclooxygenase-1 (COX-1), cyclooxygenase-2 (COX-2), and 5-lipoxygenase catalytic activity, and consequent suppression of prostaglandins, thromboxanes, and leukotriene formation [11]. It has been observed how this compound attenuated macrophage/mast cell-derived proinflammatory factors such as platelet-activating factor (PAF), tumor necrosis factor-α (TNF-α), and histamine [12]. In addition, RSV is able to inhibit chemotactic factors formation and platelet aggregation [13] supporting cardioprotective effects [14]. It has been reported that it determines an increase in the expression of endothelium nitric synthase (eNOS) and in the synthesis of nitric oxide (NO) restoring the endothelial dysfunction [15]. Furthermore, RSV has neuroprotective activity, as evidenced by its capability to improve cellular stress resistance and longevity, by increasing the activity of SIRT1 [16]: a member of the sirtuins family, comprising proteins that possess either mono-ADP-ribosyltransferase or deacetylase activity [17]. Moreover, it has been observed that RSV-induced SIRT1 activity, through disruption of the Toll-like receptor 4/nuclear factor κ-light-chain enhancer of activated B cells/signal transducer and activator of transcription (TLR4/NF-κB/STAT) signaling, reduces cytokines production in activated microglia [18]. In particular, RSV displays important neuroprotective effects on animal models with Parkinson’s disease and prevents free radical-mediated damage of neuronal cells through the activation of SIRT1 pathway [19].

On the other hand, it has been demonstrated that RSV possesses cancer chemopreventive and chemotherapeutic activity [4]. RSV chemopreventive effects are attributed mainly to the inhibition of cyclooxygenases [20,21], NF-κB [22], kinases such as protein kinase C [23] or reduced cytochrome P450, family 1, and member A1 and B1 (CYP1A1 and CYP1B1) gene expression [24]. CYP1A1 and CYP1B1 genes encode for enzymes that play a central role in metabolic activation of several procarcinogens and in the detoxification from different xenobiotic compounds [25]. Antitumor properties of RSV were demonstrated in vitro in several tumors [26] including lymphoblastic leukemia [27], colon [28], pancreatic [29], melanoma [30], gastric [31], cervical [32], ovarian [33], endometrial [34], liver [35], prostate [36], and breast [37]. These properties are mainly due to its proapoptotic and antiproliferative actions. Furthermore, RSV increases the efficacy of traditional chemotherapy and radiotherapy decreasing resistance and sensitizing tumor cells to chemotherapeutic agents [38]. Moreover, preclinical in vivo studies [39] and clinical trials confirmed its relevant antitumor actions [38,40,41,42]. 

## 2. Pharmacokinetic Characteristics of RSV

Although several reports confirmed that RSV possesses health beneficial effects, this compound shows peculiar pharmacokinetic characteristics that limit its use. In mammals, RSV is extensively metabolized and rapidly eliminated and therefore it shows a poor bioavailability [43,44]. After oral administration, RSV is absorbed at the intestinal level by passive diffusion or by membrane transporters and then released in the bloodstream where it can be detected as unmodified or metabolized molecule [45]. In fact, in the intestine, this compound undergoes a presystemic metabolism through first-pass glucuronidation and sulfate conjugation of the phenolic groups and hydrogenation of the aliphatic double bond [45]. In the bloodstream, RSV can bind to albumin and lipoproteins, such as LDL, thus forming complexes which, in turn, can be dissociated at the cellular membrane where albumin and LDL interact with the relative receptors allowing RSV entrance into cells [46]. Phase II metabolism of RSV and its metabolites occurs in the liver. Five different metabolites were detected in the urine: RSV monosulfate, two isomeric forms of RSV monoglucuronide, monosulfate dihydro-RSV, and monoglucuronide dihydro-RSV [45,47]. It has been reported that the majority of plasma RSV metabolites are RSV-3-*O*-sulfate, RSV-4′-*O*-glucuronide, and RSV-3-*O*-glucuronide, all with very little bioactivity, even if RSV-3-*O*-sulfate possesses estrogen receptor α-preferential antagonistic activity [48]. Moreover, extremely rapid sulfate conjugation by the intestine/liver appears to be the rate-limiting step in RSV bioavailability [44]. It has been demonstrated that both sulfates and glucuronides can be converted to RSV in target tissues such as liver [49]. In addition, RSV metabolites undergo enterohepatic recirculation, which allows its deconjugation in the small intestine and reabsorption [50]. 

Although RSV is quickly metabolized, oral administration is the preferred and only viable route, except for topical application. It is known that plasma concentration of the unchanged RSV depends on the dosages ingested. Several preclinical studies aimed to determine the appropriate RSV oral dosage and bioavailability in humans [51,52]. It has been demonstrated that oral dose of 25 mg of RSV resulted in plasma concentration for unchanged RSV in the range of 1 to 5 ng/mL [44]. Administration of higher doses (up to 5 g) led to the increase of unchanged RSV up to 530 ng/mL, indicating how after a high RSV dose only a low amount of the unchanged RSV is present in the plasma [53]. Even if RSV seems to be well tolerated and safe, administration of higher oral doses does not allow to improve therapeutic effects [53], but, instead, may be the cause of the side effects observed at the dose of 1 g/kg (body weight) including diarrhea, nausea, and abdominal pain [51]. Therefore, based on the findings from clinical studies, it appears that the main obstacle that must be overcome to consider RSV as a therapeutic agent is its low bioavailability [54]. For this reason, the researches focused on improving pharmacokinetic profile of RSV.

## 3. Methodological Approaches to Improve RSV Oral Bioavailability

The Biopharmaceutics Classification System (BCS), as defined by Amidon et al. [55], inserts RSV in the second class of drugs characterized by low water solubility (~30 mg/L) and high intestinal membrane permeability [55,56]. This classification provides a theoretical basis to correlate in vitro drug dissolution and in vivo bioavailability. Since RSV has a limited dissolution rate in the aqueous environment, a small increase in solubility may significantly enhance its bioavailability [57]. In this regard, a delivery system that can facilitate rapid absorption of a large amount of RSV, could effectively increase its plasma concentration. In the last decade, in order to improve RSV’s poor bioavailability, various methodological approaches have been developed. These include several delivery systems such as the RSV encapsulation in lipid nanocarriers or liposomes, emulsions, micelles, insertion into polymeric nanoparticles, solid dispersions, and nanocrystals (Figure 2). 

Encapsulation offers a potential approach for enhancing the RSV solubility, stabilizing it against *trans*-to-*cis* isomerization and improving its bioavailability. This approach may potentially be used to protect RSV from degradation and to control its release when orally administered [57]. Solid dispersion drug delivery systems are attractive as alternative solubilization methods because manipulating the carriers and powder properties of the active components can improve the solubility, dissolution, and even in vivo absorption of the active components [58,59,60]. Furthermore, nanonization of the drug particles producing nanocrystals represents a very promising strategy that improves solubility, dissolution rate of insoluble drugs, physical and chemical stability, compatibility in oral forms of dosage, and oral bioavailability [61].

### 3.1. Lipid Nanocarriers and Liposomes

In order to increase the intestinal uptake of hydrophobic RSV, many lipid-based devices were formulated. Nanoencapsulation of specific compounds in lipid nanocarriers or liposomes represents a good strategy to significantly increase aqueous solubility and chemical stability. It has been demonstrated that lipid nanoparticles act as a vehicle to enhance the oral bioavailability and therapeutic potential of RSV [62,63] (Table 2). 

Solid lipid nanoparticles (SLNs) and nanostructured lipid carriers (NLCs) have been found to protect the incorporated RSV from rapid metabolism, to increase its physical stability and to allow a controlled release after uptake [62].

In the lipid-based encapsulation system, the presence of lipid itself could enhance the absorption of lipophilic compound [64]. In particular, Basavaraj et al. demonstrated how RSV encapsulation in a liposomal vehicle composed of phosphatidyl choline increases oral absorption of the native unmetabolized compound [65]. Recently, biocompatible and biodegradable *trans*-RSV encapsulated in lipid nanocarriers (RSV-nano) or liposomes (RSV-lipo) were synthesized and their cytotoxicity and ability to release RSV in the cellular compartment were evaluated in 3T3-L1 fibroblasts [66]. Results confirmed that RSV-lipo have higher physical and chemical stability than RSV-nano, which, however, displays a more prolonged release than RSV-lipo. Both RSV-nano and RSV-lipo increased cellular RSV content in 3T3-L1 cells, but RSV-lipo has better biological activity due to its higher physical and chemical stability at room temperature [66]. Overall, this study indicated how the lipid-based encapsulation system can represent a good strategy to increase some pharmacokinetic parameters of RSV with minimal side effects. 

### 3.2. Nanoemulsions

Different oil/water (O/W) nanoemulsion-based delivery systems have been developed in order to optimize the bioavailability of encapsulated RSV for potential oral administration (Table 2). Sessa et al. demonstrated the ability of subcellular size nanoemulsions, produced by high pressure homogenization (HPH), to protect RSV from chemical degradation, preventing oxidation reactions that would lead to conversion into *cis*-form [67]. In another work, the same author, using human carcinoma intestinal Caco-2 cell monolayers as model systems to simulate the intestinal epithelium, revealed that nanoemulsions encapsulating RSV lack cytotoxicity. Furthermore, these formulations, particularly the lecithin-based nanoemulsions, allow RSV delivery through cell monolayers avoiding its degradation and ensuring sustained release [68]. It has been reported how the self-nanoemulsifying drug delivery systems (SNEDDS) are excellent for delivering lipophilic compounds, based on good stability and improved rate and extent of absorption. Yen et al. demonstrated how the SNEDDS for RSV not only enhance its oral bioavailability but may also exert antifatigue effects in rats [69]. In particular, pharmacokinetic studies revealed that oral bioavailability of the optimized RSV-SNEDDS increased by 3.2-fold compared with the unformulated RSV-solution. The UDP-glucuronosyltransferase (UGT) inhibitory excipient-based self-microemulsion (SME) represents a different system to increase RSV oral bioavailability. In a recent study, a novel RSV-loaded SME using UGT inhibitory excipients was prepared to inhibit intestinal metabolism [70]. Toxicity studies demonstrated that SME systems, containing excipients with and without inhibitory activities, named SME1 and SME2, respectively, displayed safety in Caco-2 cells. Moreover, both SME1 and SME2 conferred significantly higher RSV uptake and sustained release than the free drug. In vivo data from rats demonstrated that SME1 system increased oral RSV bioavailability compared to both RSV-free and SME2 through modulating the glucuronidation by UGT inhibitory excipients [70]. 

### 3.3. Micelles

It has been reported that micellar solutions of bile acids can solubilize poorly soluble organic compounds, improving their resorption [71] (Table 2). Atanacković et al. examined the ability of different bile acids micellar solutions to make RSV more soluble. In addition, they evaluated cell membrane toxicity degree by measuring the hemolytic potential [72]. Results from this study showed that among the tested bile acids micellar solutions, those containing 3,7,12-triketocholic acids displayed the smallest membranolytical potential and solubilized RSV more effectively [72]. A recent study investigated the effect of micellar solubilization on the bioavailability of vineatrol30—a standardized ethanolic extract of grapevine shoots, which contains RSV as well as considerable amounts of RSV oligomers [73]. The study was carried out on twelve healthy volunteers (six women, six men) randomly divided into two groups: one administered with a single dose of vineatrol30 in a powder formulation, the other administered with vineatrol30 as a liquid micellar solution. The results showed that liquid micellar solubilization significantly increased the oral bioavailability of RSV from vineatrol30 when compared to the native powder, without producing side effects [73]. 

Another recent study [74] revealed that highly hydrated colloidal particles, such as β-casein micelles, represent good oral carriers. Casein micellar-like forms are very effective in the encapsulation of hydrophobic compounds [75]. In particular, it has been demonstrated that when casein nanoparticles are administered orally to rats and reach gut epithelium surface they control the RSV release rate and ameliorate its absorption and oral bioavailability (Table 2) [74]. Moreover, in the presence of proteins, RSV would be more protected from *trans*-to-*cis* isomerization than in the free-form [76]. Results obtained from this study are very interesting because demonstrated a good correlation between in vitro (release of RSV from nanoparticles) and in vivo (RSV plasmatic concentration) data. 

### 3.4. Polymeric Nanoparticles

Among various nanoparticles (NPs)-based formulations prepared to enhance RSV delivery some involve encapsulation into biodegradable polymeric NPs [77]. Poly(lactic-co-glycolic acid) (PLGA) is used as the hydrophobic portion of polymeric NPs for a variety of biomedical delivery systems and has the advantage of being biodegradable and biocompatible [78]. Recently, it has been demonstrated that RSV-PLGA-NPs represent a stable drug delivery method because it is characterized by small particle size, high capsulation efficiency, well-controlled drug release, enhanced chemical stability, water solubility, and bioactivity (Table 2) [79]. The biological effects of polymeric PLGA nanoparticles encapsulating RSV have been also evaluated in LNCaP prostate cancer cell line [80]. The study revealed that nanoparticles encapsulating RSV exerted greater cytotoxicity compared to free RSV, at all tested concentrations. In addition, nanoparticles presented no cytotoxic effects on murine macrophages suggesting the potential use of these RSV formulations for prostate cancer chemoprevention and chemotherapy, without adverse effect on normal cells [80]. In another recent work, oral bioavailability and the in vitro anti-inflammatory activity of RSV-loaded galactosylated PLGA NPs (RSV-GNPs) have been investigated in rats and in lipopolysaccharide-induced RAW 264.7 macrophage cell line, respectively [81]. After oral administration, RSV-GNPs showed better oral bioavailability compared to RSV. Similarly, in lipopolysaccharide-induced RAW 264.7 cells, RSV-GNPs displayed greater anti-inflammatory activity [81]. These results confirmed that RSV-GNPs, increasing RSV intestinal absorption and improving its bioavailability, might represent a valid therapeutic approach against inflammatory disease. 

### 3.5. Solid Dispersions

It has been reported how RSV solubility can also be enhanced by preparing a solid dispersion of RSV on magnesium dihydroxide (RSV@MDH). Particularly, in a recent work, Spogli et al., using the dissolution test in simulated gastric environment, demonstrated how RSV@MDH had higher solubility compared to RSV alone. In addition, the oral administration of 50 mg/kg of RSV from RSV@MDH in rabbits determined a 3-fold increase in RSV bioavailability (Table 2) [59]. Similar results were obtained by incorporation of grape peel extract (GPE) into a solid dispersion delivery system or dripping pill (DP) (GPEDP) oral delivery system and testing effects on rats [58]. Pharmacokinetic profile analyses displayed increased RSV bioavailability more than 10-fold, suggesting how the DP oral delivery system can represent a valid device for clinical applications [58].

### 3.6. Nanocrystals

Singh et al., using the probe sonication method, demonstrated how drug nanonization can represent a valid approach in preparing nanocrystals (NCs) with physical and chemical stability, higher dissolution profile, and then in vivo-enhanced oral bioavailability compared to RSV (Table 2) [61]. Particularly, the in vitro cellular efficacy against MDA-MB-231 breast cancer cells and in vivo pharmacokinetic parameters of stable *trans*-RSV nanocrystals (t-RSV-NCs) have been evaluated [61]. In MDA-MB-231 cells, t-RSV-NCs displayed improved cytotoxicity compared to *trans*-RSV in a concentration- and time-dependent manner. Furthermore, t-RSV-NCs were more effective in inhibiting cell cycle arrest at the S phase and in inducing apoptosis than RSV [61]. In vivo pharmacokinetic studies performed in Sprague-Dawley rats using *trans*-RSV and t-RSV-NCs, revealed that the plasma concentration profile of t-RSV-NCs is enhanced compared to *trans*-RSV [61]. 

### 3.7. Limitations and Similarities of Different Methodological Approaches 

As above-discussed, different strategies to increase RSV solubility have been developed. In vitro studies demonstrated that the increase in RSV solubility determines a partial saturation of its metabolism with consequent improvement of its bioavailability [82]. Growing evidences revealed that encapsulation of RSV into solid lipid nanoparticles, liposomes, emulsions, or micelles, or insertion into polymeric nanoparticles improves RSV absorption and stability [63,83]. However, the lipid-based formulations showed several disadvantages. For instance, many have low solvent capacity, unless the active components are highly lipophilic, and low stability of active components loaded in liposomal and nanoparticle systems [84]. Furthermore, drug encapsulation in a lipid matrix reduces the drug load capacity and, at the same time, increases its amount to achieve the therapeutic results desired [84]. Concerning to the use of nanoparticles as carriers, they have several drawbacks including the ability to cross biological membranes, such as the blood–brain barrier, and any modification has to be carefully evaluated because it could reduce their half-life due to the response of immune system in the liver and spleen [85]. Solid dispersion drug delivery systems are attractive as alternative solubilization methods because manipulating the carriers and powder properties of the active components can improve the solubility, dissolution, and even in vivo absorption [60]. The bioavailability can be also improved by controlling molecular weight, carrier composition or crystalline status, and powder porosity of the active components [86]. However, the therapeutic application of the solid dispersions is limited by the thermodynamic instability of the molecules in the amorphous state which leads to the drug uncontrolled crystallization during storage. Compared to the above-mentioned methodological approaches, drug nanonization to produce stable nanocrystals (NCs) represents a very promising strategy that is able to improve the solubility and dissolution rate of insoluble drugs in water, physical and chemical stability, compatibility in oral forms of dosage and, finally, in vivo oral bioavailability [61]. In addition, NCs present the advantages of simplest composition, lower manufacturing cost, and excipient side effects [61,87]. However, due to the extremely high free surface energies, NCs undergo agglomeration; for this reason they require the use of stabilizers that can be adsorbed on the particle surface to generate repulsive forces leading to the steric or electrostatic stabilization of formulation. 

## 4. RSV Derivatives

Clinical trials emphasized potential drawbacks of RSV poor bioavailability [54,88]. Currently, many studies have been performed to synthesize new and more powerful RSV analogs that display better pharmacokinetic properties together with better biological activity than RSV. Several synthetic approaches have been carried out to produce RSV derivatives structurally different for type, number, and positions of substituents [89,90,91] (Figure 3). The presence of three free hydroxyl groups in the backbone of RSV makes it susceptible to extensive phase-II conjugation reactions in vivo, which confers its poor bioavailability [88].

In the last years, methoxylated, hydroxylated, and halogenated RSV derivatives received more attention for their beneficial biological effects and potential increased oral bioavailability [91] (Table 3). Some of these possess a stronger pharmacological potency and have a better pharmacokinetic profile than RSV itself.

Methoxylation increased metabolic stability and also the time length required to reach the plasma concentration peak. Therefore, 3,4,5,4′-tetramethoxystilbene, administrated to mice, showed more favorable pharmacokinetic properties than RSV, because it yielded higher levels of drug in the small intestine, colon mucosae, and brain [92]. However, the oral bioavailability of the methoxylated compounds depends on the positions of the methoxylic group [93]. In general, polyhydroxylated derivatives containing less than three hydroxylic groups on the stilbene moiety exhibit very poor oral bioavailability [94]. On the other hand, *trans*-3,5,2′,4′-tetrahydroxystilbene showed a faster absorption rate than RSV because of better solubility due to the presence of one additional hydroxyl group [95]. Furthermore, the presence of halogen groups to the stilbene moiety, increases compound lipophilicity and improves the oral bioavailability and therapeutic application of these compounds [96,97].

### 4.1. Methoxylated RSV Derivatives

Several reports indicate that substitution of hydroxylic with methoxylic groups on RSV potentiates biological activity [98], improving antiproliferative and proapoptotic effects of RSV analogs [99]. Pharmacokinetic studies revealed that the presence of two methoxy groups in the stilbene core, to obtain the *trans*-3,5-dimethoxy-4′-hydroxystilbene (pterostilbene) (**1**) (Table 3), increases its lipophilicity, absorption, cellular uptake, and oral bioavailability compared to RSV [100,101]. Pterostilbene displays antioxidant, cardioprotective, neuroprotective, antidiabetic, and antitumor properties [98]. The antioxidant effects of pterostilbene are attributed to its unique structure with one hydroxylic group and two methoxylic groups that may scavenge ROS [102] including hydrogen peroxide (H_2_O_2_) and the superoxide anion, which are implicated in the initiation and pathogenesis of several diseases [103]. Oral administration of pterostilbene scavenges free radicals and decreases superoxide dismutase, catalase, glutathione peroxidase, and glutathione-S-transferase activity in the liver and kidney of streptozotocin–nicotinamide-induced diabetic rats [104]. In the same in vivo model, it exerts beneficial effects on glucose concentration decreasing plasma glucose and gluconeogenic enzyme activity as well as increasing plasma insulin levels [105]. Pterostilbene possesses cardioprotective effects [106], attenuating the ischemia/reperfusion-induced inflammatory reaction in the rat heart and reducing hypoxia-reoxygenation injury in cardiomyocytes by SIRT1 activation [107]. In addition, pterostilbene protects Sprague-Dawley rats against diabetic myocardial ischemia-reperfusion (IR) injury reducing oxidative stress and apoptosis via increased phosphorylation of adenosine monophosphate-activated protein kinase (AMPK) [108]. Moreover, it decreases cardiac oxidative stress in fructose-fed rats through stimulation of AMPK/nuclear factor erythroid 2-related factor 2 (Nrf2)/Heme-oxygenase (HO-1) pathway [109]. Pterostilbene also possesses neuroprotective proprieties restoring cognitive function during aging process [110] and in Alzheimer’s disease [111]. In fact, using the BV-2 murine microglia cell line, protection from amyloid-β-induced inflammation through inhibition of NACHT, LRR, and PYD domains-containing protein 3 (NLRP3)/caspase-1 inflammasome pathway was demonstrated [111]. Moreover, this compound inhibits growth of different human cancers, including pancreatic [112], lung [113], epatic [114], colon [115], gastric [116], melanoma [117], prostate [118], and breast [119]. 

Trimethoxystilbene, also called *trans*-3,4′,5-trimethoxystilbene (**2**) (Table 3), is a methoxylated RSV derivative that displays a better cardioprotective [120], anti-Hepatitis C Virus (HCV) [121], and antitumor [122] activity than RSV. It has been demonstrated that trimethoxystilbene protects Chinese hamster ovary (CHO) cells from DNA damage induced by ionizing radiation (IR), and reduces cell proliferation, delaying mitosis progression by increasing the number of cells in metaphase [123]. Moreover, trimethoxystilbene, unlike RSV and pterostilbene, reduces cell growth, alters microtubule polymerization, and induces mitotic catastrophe in HeLa cervical cancer cells [124].

*Trans*-3,4,5,4′-tetramethoxystilbene (DMU 212 or TMS) (**3**) (Table 3) is another methoxylated RSV derivative that exhibited more favorable pharmacokinetic properties than RSV. This compound administrated in mice, is present at higher levels in the small intestine, colon mucosae, and brain compared to RSV [92]. In addition, DMU 212 appears to be a slightly more potent growth inhibitor than RSV in human colon cancer cell lines HT-29 and HCA-7 [92] and DLD-1 and LOVO [125]. Several reports indicated that 3,4,5,4′-tetramethoxystilbene also possesses stronger bioactivity than RSV in suppressing prostate [126], ovarian [127], non-small cell lung [128], breast [129], and liver [130] tumors. 3,4,5,4′-Tetramethoxystilbene is found to be a strong inhibitor of human cytochrome P450 1B1 (CYP1B1) [131] that is overexpressed in a variety of hormone-dependent human cancers [132,180]. CYP1B1 activates many environmental mutagens [181] and also catalyzes the 4-hydroxylation of estrogens [182], considered to be an important step in hormonal tumorigenesis. By suppressing CYP1B1 expression, 3,4,5,4′-tetramethoxystilbene inhibits human mammary cancer cell growth [132]. 

CYP1B1 expression can be reduced by *trans*-2,4,3′,4′,5′-pentamethoxystilbene (PMS) (**4**) [132] (Table 3) which additionally decreases CYP1A1 expression [133], another enzyme associated to carcinogenesis [183]. Similarly to TMS, PMS exerts better inhibitory effects on cancer cell growth compared to RSV, particularly in breast [134] and colon tumors [135]. However, pharmacokinetics studies performed in Sprague-Dawley rats indicated that PMS displays lower oral bioavailability than other methoxylated stilbenes due to the presence of the methoxylic group at the two position [136].

### 4.2. Hydroxylated RSV Derivatives

It was reported that the introduction of additional hydroxylic groups significantly increased the biological activity of RSV analogs [168,169,170,184]. Ovesná et al. demonstrated that *trans*-stilbene compounds having 4-hydroxy group, a double bond, and bearing ortho-diphenoxyl or para-diphenoxyl functionalities display higher chemopreventive activity than *trans*-RSV [184]. Additionally, Chen et al., using liquid chromatography-tandem mass spectrometry, determined the pharmacokinetic profiles of *trans*-4,4′-dihydroxystilbene (DHS) (**5**) (Table 3) in Sprague-Dawley rats. The study revealed that DHS is absorbed slowly and possesses low oral bioavailability; however, when DHS is solubilized in hydroxypropyl-β-cyclodextrin it is absorbed rapidly and displays improved bioavailability [94]. Similarly to RSV, DHS exerts cardioprotective activity by inhibiting both secretion and mRNA expression of endothelin-1, a vascular tension regulator, and by decreasing mRNA levels of endothelin-converting enzyme-1, a protein involved in the proteolytic processing of endothelin-1 [137]. Although DHS shows antioxidant effects against LDL peroxidation induced by free radicals [138], it can also act as a prooxidant compound depending on the cell type. DHS protects from H_2_O_2_-induced apoptotic death in the pheochromocytoma PC12 cell line, while increases ROS production and activates caspase-3–depending apoptosis in the human neuroblastoma SHSY-5Y cell line [139]. DHS possesses better tumor growth inhibitory potential compared to RSV. Recently, it has been demonstrated that DHS inhibits DNA replication via its ability to interact with ribonucleotide reductase regulatory subunit M2 (RRM2) and decreases tumor growth of a wide spectrum of cancer cell lines [140]. DHS suppresses melanoma tumor growth and metastases to the lungs [141] and reduces human neuroblastoma tumor growth by mitochondrial and lysosomal damages [142]. Moreover, it inhibits normal 3T3 mouse fibroblasts transformation and suppresses more efficiently both anchorage-dependent and -independent MCF-7 human breast cancer cell growth [143]. 

*Trans*-3′,4′,3,5-tetrahydroxy-stilbene, also called piceatannol (**6**) (Table 3), is another hydroxylated RSV derivative that displays biological functions similar to RSV [144]. Setoguchi et al. evaluated metabolism and absorption of piceatannol in rats after intragastric administration [145]. The study revealed that piceatannol undergoes more complex metabolism than RSV involving glucuronidation, sulfation, and methylation reactions. In fact, piceatannol metabolites are piceatannol-mono and diglucuronide, *O*-methyl piceatannol-monoglucuronide, *O*-methyl piceatannol-monosulfate Moreover, while the AUC for the intact piceatannol is higher than intact RSV, in the case of the AUC for total (intact and metabolites) piceatannol, the value is lower than what seen for total RSV [145]. In order to improve piceatannol bioavailability, several strategies have been developed. Prenylated forms of piceatannol showed slower glucuronidation and higher biological activity compared to RSV [146]. Delivery systems can also increase piceatannol bioavailability. The use of polymeric nanoparticles could facilitate their transport and undamaged delivery to the gastrointestinal tract following oral administration [147]. Thus, chitosan/poly(lactic acid)-coated piceatannol nanoparticles [148] and piceatannol complexed with β- [149] or α- cyclodextrin– [147] improve in vitro drug release, solubility and stability, and in vivo absorption, respectively. 

It has been demonstrated that piceatannol possesses cardioprotective [150], neuroprotective [151], and chemopreventive/therapeutic activities [152]. A recent work demonstrated that piceatannol protects H9c2 cardiomyocytes against oxidative stress, cytotoxicity, and H_2_O_2_-induced apoptosis [150]. It is known that in ischemia/reperfusion injury, while eNOS expression is downregulated, iNOS is upregulated [153]. In cardiac injury, eNOS has been implicated into PI3K/Akt/eNOS pathway and iNOS has been implicated into NF-κB/iNOS/NO pathway in response to cellular stress [154]. In H9c2 cardiomyocytes, piceatannol upregulated PI3K, p-Akt, and eNOS activities and downregulated iNOS expression [150].

Piceatannol is estimated as promising candidate for the treatment of age-associated diseases. In vivo studies performed in mice, revealed that, after chronic injection of d-galactose (d-gal), which accelerates the aging process, piceatannol allowed the maintenance of spontaneous motor activity and enhanced spatial learning and memory abilities [151]. In addition, a morphometric analysis showed how piceatannol prevents d-gal-induced neuronal loss, increases the number of Nissl bodies, and promotes cell proliferation in the hippocampus and cortex [151]. These effects are dependent on activation of Nuclear factor erythroid 2-related factor 2 (Nrf2), a transcription factor that regulates gene expression of a wide variety of antioxidant cytoprotective and phase II detoxification enzymes [151]. In another work, it has been demonstrate that piceatannol protects PC12 rat pheochromocytoma cells from oxidative stress, blocking beta-amyloid (Aβ)-induced ROS accumulation [155].

Several reports indicated that piceatannol is an antitumor agent for its ability to inhibit growth, invasion and metastasis and to induce apoptosis of a wide range of cancer cells [152]. Chemoterapeutic effects of this compound are evaluated in leukemia [156], breast [157], bladder [158], prostate [159,160], melanoma [161], hepatic [162], and colon [163,164] cancer. Piceatannol treatment determined cancer cell cycle arrest at G0/G1 [158] and S phases [164], and additionally apoptotic effects through both extrinsic and intrinsic apoptotic pathways [144,165]. In fact, piceatannol causes cell cycle arrest, growth inhibition, and apoptosis in human intestinal Caco-2 cells. These effects are related to its antioxidant capacity and ability to block the arachidonic acid cascade [164]. In several leukemia cell lines (THP-1, HL-60, U937, and K562) piceatannol induced apoptosis, downregulating X-linked inhibitor of apoptosis protein (XIAP) expression, regardless of the inhibition of ROS generation [156]. Kim et al. reported that in androgen-insensitive DU145 prostate cancer cells, piceatannol determines apoptosis by increased cleavage of caspase-8, -9, -7, and -3; poly(ADP-ribose) polymerase (PARP); cytochrome c release; and increased Bid, Bax, Bik, Bok, and Fas protein expression [165]. Piceatannol is also able to reduce cancer invasion and metastasis. In MDA-MB-231 breast cancer cells inhibition of invasion, migration, and adhesion was related to decreased matrix metalloproteinase-9 (MMP-9) activity and expression [157]. These effects are dependent on phosphatase and tensin homologue (PTEN) increase, PI3K/AKT pathway downregulation and NF-κB inhibition [157]. The same results are obtained in DU145 prostate cancer cells where piceatannol inhibited tumor necrosis factor-α (TNF-α)-induced invasion by suppressing MMP-9 activation via the AKT-mediated NF-κB pathway [159].

Among polyhydroxylated stilbenes, the 3,3′,4,5,5′-pentahydroxy-*trans*-stilbene (PHS) (**7**) (Table 3) demonstrated high activity on cancer cell growth. PHS suppresses HT-29 human colorectal carcinoma cell proliferation through oxidative stress-mediated apoptosis [166] and inhibits 12-O-tetradecanoylphorbol-13-acetate (TPA)-induced neoplastic transformation in JB6 P+ mouse epidermal cells [167].

Another hydroxylated RSV derivative that has been extensively studied is 3,3′,4,4′,5,5′-hexahydroxystilbene (**8**) (Table 3). This compound displays a potent antiviral activity against several human immunodeficiency virus (HIV) variants via blocking viral attachment to host cells [171], and exerts antioxidant [169] and antitumor actions in promyelocytic leukemia cells [168] more effectively than RSV. Additionally, it has been reported that 3,3′,4,4′,5,5′-hexahydroxy-*trans*-stilbene represents a new drug for the treatment of metastatic melanoma [172]. Evaluating the effects of this compound on 1205Lu metastatic melanoma cell line, primary human melanoma cells (MCM1, MCM19, and MLNM1) and MKN28 gastric cancer cell line, it interferes with several pathways involved in apoptosis, cell cycle regulation, DNA damage repair, and cell migration and metastasis [172]. In addition, the authors confirmed in vitro data in an in vivo model, demonstrating how it decreases tumor growth and prevents metastatic spread of melanoma cells to distant lymph nodes [172]. The antitumoral effects of 3,3′,4,4′,5,5′-hexahydroxystilbene were confirmed in ZR-75-1, MDA-MB-231, and T47D human breast cancer cells [170]. The authors demonstrated how this derivative caused cell growth inhibition and apoptosis via caspase-8 activation in MDA-MB-231 cells, and caspase-9 and caspase-3 activation in all three tested cell lines [170]. In particular, caspase activation is associated with p53 increase and mitochondrial superoxide dismutase downregulation [170]. 

### 4.3. Halogenated Derivatives

Li et al. demonstrated that halogenation of RSV on one of the two aromatic rings, produces more lipophilic derivatives with better antihemolysis and antimicrobial activity than RSV [97]. In particular, the authors, using the 2,2′-azobis (2-amidinopropane hydrochloride) (AAPH)-induced human red blood cells (RBC) hemolysis model, and evaluated the antioxidant activity of several chlorinated and brominated RSV derivatives. The results indicated that all the used halogenated derivatives are more lipophilic and effective at inhibiting RBC hemolysis than RSV [97]. In addition, the introduction of chlorine or bromine atoms on RSV improves in vitro antimicrobial activity against Gram-positive bacteria (*S. aureus*), Gram-negative bacteria (*E. coli*), and yeast (*C. albicans*) [97]. Using a synthetic strategy, a series of 2,3-thiazolidin-4-one RSV derivatives (**9**) (Table 3) were synthesized with an azetidin-2-one nucleus connected to two aromatic rings [173,174]. Cell viability in response to 3-chloro-azetidin-2-one RSV derivatives was investigated on human breast cancer cell lines [174]. 1,4-Bis(4-bromophenyl)-3-chloroazetidin-2-one and 3-chloro-4-(4-chlorophenyl)-1-(4-iodophenyl)azetidin-2-one displayed inhibitory effects in a dose-dependent manner in estrogen receptor positive (ER+) MCF-7 and in estrogen receptor negative (ER-) SKBR3 cell lines [174]. Several fluorinated RSV derivatives were tested on lung cancer and melanoma cell lines. Among them, the (*E*)-3,5-difluoro-4′-acetoxystilbene (**10**) (Table 3), has a greater antiproliferative action [96]. In addition, evaluating the effects of this compound on other cancer cell lines, the authors revealed how it has a better anticancer activity against leukemia, colon, lung, breast, melanoma, prostate, ovarian, central nervous system, and renal cancers [96]. More effective than RSV on the inhibition of human lung carcinoma A549 cell growth was the 3,4,5-trimethoxy-4′-bromo-*cis*-stilbene (**11**) (Table 3) [175]. Particularly, this compound determined cell cycle arrest at the G2/M phase and induced apoptosis as revealed by upregulation of p53 expression and cytochrome c release [175].

### 4.4. Other RSV Derivatives

Using a chemical strategy that replaces the alkene linker between the two aromatic rings with a heterocyclic system [173], rigid derivatives structurally correlated to 2,3-thiazolidin-4-ones (**12**) (Table 3) were synthesized that displayed higher cytotoxic activity. They manifested higher ability to inhibit in vitro cancer cell growth and, in some cases, a greater bioaccessibility than RSV analog [176]. Some of these compounds showed strong inhibitory effects on MCF-7 and SKBR3 cell growth. Particularly, 2-(3,4-dimethoxyphenyl)-3-(4-hydroxyphenyl)-thiazolidin-4-one, 2-(4-chlorophenyl)-3-(4-hydroxyphenyl) thiazolidin-4-one, 2,3-diphenylthiazolidin-4-one, 2-(naphthalen-1-yl)-3-*p*-tolylthiazolidin-4-one, and 2-(3,4,5-trimethoxyphenyl)-3-*p*-tolylthiazolidin-4-one displayed potent cytotoxic activity against MCF-7 and SKBR3 cells, suggesting how the biological action of these molecules could also be influenced by the different estrogenic receptor status [176]. In particular, in MCF-7 cells, 2-(3,4-dimethoxyphenyl)-3-(4-hydroxyphenyl)-thiazolidin-4-one and 2-(4-chlorophenyl)-3-(4-hydroxyphenyl)-thiazolidin-4-one could interfere with the estrogen receptor α (ERα)-dependent pathway, while in SKBR3 cells, 2,3-diphenylthiazolidin-4-one, 2-(naphthalen-1-yl)-3-*p*-tolylthiazolidin-4-one, and 2-(3,4,5-trimethoxyphenyl)-3-*p*-tolylthiazolidin-4-one could antagonize the alternative estrogen receptor G protein-coupled estrogen receptor (GPER) [176], which is known to be involved in estradiol-dependent SKBR3 cell growth [185,186].

The anticancer activity of 4-(6-hydroxy-2-naphthalen-2-yl)-1,3-benzenediol (HS-1793) (**13**) (Table 3) RSV derivative has been evaluated in FM3 murine breast cancer cells [177]. In this cell type, HS-1793 caused inhibition of cell proliferation, followed by apoptosis, at doses (3–25 μM) lower than RSV (300 μM) [177]. Antiproliferative and apoptotic effects of HS-1793 have been confirmed in MCF-7 and MDA-MB-231 cells [178]. Particularly, the authors showed that this compound induced p53/p21WAF1/CIP1-dependent apoptosis in MCF-7 cells, whereas it determined p53-independent apoptosis in MDA-MB-231 cells [178]. Recently, a series of isoprenylation RSV dimer derivatives was prepared and their biological activities were evaluated against Alzheimer’s disease [179]. Some of these compounds inhibited human monoamine oxidase B (hMAO-B) and exerted good anti-inflammatory and antioxidant activities. In addition, 5-((E)-2-(3-(3,5-dihydroxy-4-(3-methylbut-2-en-1-yl)phenyl)-2-(4-hydroxyphenyl)-2,3-dihydrobenzofuran-5-yl)vinyl)-2-(3-methylbut-2-en-1-yl)benzene-1,3-diol) (**14**) (Table 3) and 5-((E)-2-(3-(5-hydroxy-2,2-dimethylchroman-7-yl)-2-(4-hydroxyphenyl)-2,3-dihydrobenzofuran-5-yl)vinyl)-2-(3-methylbut-2-en-1-yl)benzene-1,3-diol (**15**) (Table 3) protected neuronal cells from LPS stimulation and H_2_O_2_-induced BV2 microglial cell inflammation, thus making these compounds potential neuroprotective agents against Alzheimer’s disease [179].

## 5. Conclusions 

RSV has gained interest as a nontoxic agent that displays multiple health beneficial effects, including antioxidant, anti-inflammatory, cardioprotective, neuroprotective, and antitumor actions. However, its rapid metabolism limits its biological effects in vivo and, consequently, its clinical applications. In the last decade, in order to improve the pharmacokinetic properties of RSV, various methodological approaches and different synthetic derivatives have been prepared. An increase in biological activity and, particularly, in RSV antitumor actions can be seen either with some of these approaches or in some new molecules. These results encourage us to perform new in vivo studies to identify the RSV delivery systems and/or RSV derivatives that can be used for clinical applications.

## Figures and Tables

**Figure 1 ijms-20-01381-f001:**
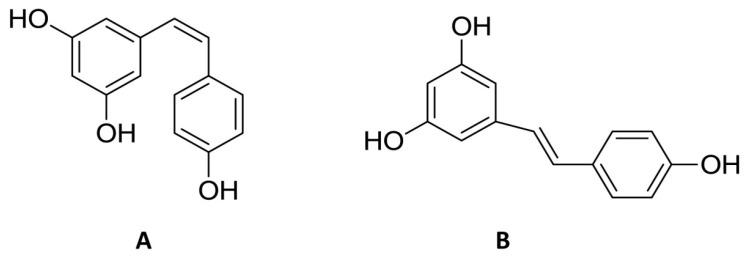
Chemical structure of *cis* (**A**) and *trans* (**B**) resveratrol.

**Figure 2 ijms-20-01381-f002:**
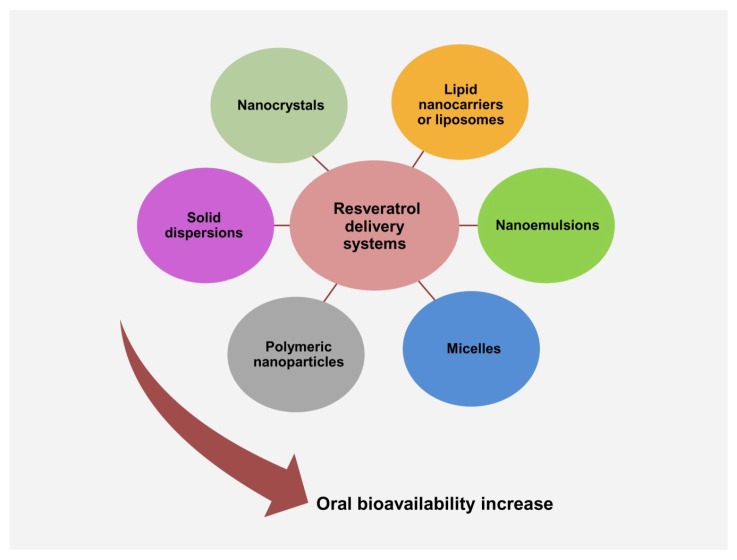
Principal resveratrol delivery systems to enhance its oral bioavailability.

**Figure 3 ijms-20-01381-f003:**
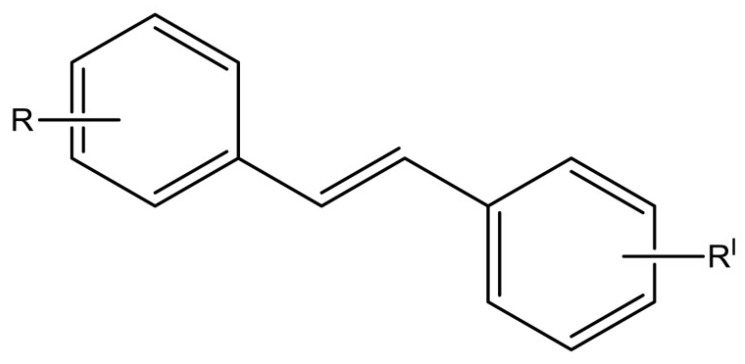
General chemical structure of resveratrol derivatives.

**Table 1 ijms-20-01381-t001:** Principal mechanisms through which resveratrol exerts its biological effects.

Biological Effects	Mechanisms	References
**Antioxidant**	Decrease of ROS and free radicals; increase of endogenous antioxidant biosynthesis	[6]
	Reduction of copper-catalyzed oxidation	[7]
	Inhibition of LDL peroxidation	[8]
	Inhibition of membrane lipids peroxidation	[9]
	Decrease of intracellular concentration of ApoB, cholesterol esters and triglycerides secretion rate	[10]
**Anti-Inflammatory**	Inhibition of COX-1, COX-2, and 5-lipoxygenase catalytic activity	[11]
	Inhibition of PAF, TNF-α, and histamine	[12]
**Cardioprotective**	Inhibition of chemotactic factors formation and platelet aggregation	[13,14]
	Increase of eNOS expression and NO synthesis	[15]
**Neuroprotective**	Increase of SIRT1 activity	[16]
	Reduction of cytokines production in activated microglia	[18]
	Prevention of free radical-mediated damage through SIRT1 pathway activation	[19]
**Antitumor**	Inhibition of	
	Cyclooxygenases	[20,21]
	NF-κB	[22]
	Kinases such as protein kinase C	[23]
	CYPA1 and CYPB1	[24]
	Apoptosis induction and proliferation inhibition in several tumors:	[26]
	Lymphoblastic leukemia	[27]
	Colon	[28]
	Pancreatic	[29]
	Melanoma	[30]
	Gastric	[31]
	Cervical	[32]
	Ovarian	[33]
	Endometrial	[34]
	Liver	[35]
	Prostate	[36]
	Breast	[37]

**Table 2 ijms-20-01381-t002:** Principal characteristics of resveratrol (RSV) delivery systems for improving its oral bioavailability. For some studies in the table the common pharmacokinetic parameters of RSV are reported. SME-1 and -2: self-microemulsion containing excipients with or without inhibitory activities, respectively; AUC: area-under-curve; C_max_: mean maximum concentration; (t1/2): elimination half-life; T_max_: time to maximum concentration.

Delivery System	Models/Methods Used	Results of Studies	References
**Lipid Nanocarriers or Liposomes**			
Solid lipid nanoparticles (SLNs) and nanostructured lipid carriers (NLCs)	In vitro release simulation studies in liquid dosage forms by cellulose dialysis bag method and in gastrointestinal transit using gastric and intestinal fluid; stability studies by measurements of particle size, polydispersity index, and zeta potential.	Prolonged release over several hours for both nanosystems. ↑ Stability.	[62]
Lipid nanocarriers (RSV-nano) or liposomes (RSV-lipo) encapsulating RSV	In vitro release study by dialysis bag method; stability studies by measurements of particle size, polydispersity index, and zeta potential; studies on 3T3-L1 mouse adipocytes cell line	↑ Solubility and stability. RSV-nano release more prolonged than RSV-lipo. Biological activity: RSV-lipo > RSV-nano. ↑ Cellular content by both RSV-nano and RSV-lipo. No cellular toxicity.	[66]
**Nanoemulsions**			
Nanoemulsions encapsulating RSV	In vitro release study by dialysis bag method; stability studies by measurements of droplet size and polydispersity index; studies on Caco-2 human intestinal cell line	↑ Solubility and stability. Sustained release. ↑ Membrane passive transport and cellular uptake. No cellular toxicity.	[68]
Self-nanoemulsifying drug delivery systems (SNEDDS)	In vitro solubility studies in different solvents; stability studies by measurements of droplet size, polydispersity index, and dispersibility test.	↑ Solubility and stability.	[69]
	In vivo studies on Sprague-Dawley rats.	Improved in rate and extent of absorption. ↑ AUC (3.3-fold) and C_max_ (2.2-fold) from SNEDDS. The t1/2 and T_max_: RSV-SNEDDS = RSV-solution groups. ↑ Oral bioavailability (3.2-fold). Antifatigue pharmacological effect in rats.	
UDP-glucuronosyltransferase (UGT) inhibitory excipient-based self-microemulsion (SME).	In vitro release study by dialysis bag method; stability studies by measurements of particle size, polydispersity index and zeta potential; studies on Caco-2 human intestinal cell line.	↑ Solubility and stability. Sustained release in SME1 and SME2. ↑ Cellular uptake and transport. No cellular toxicity.	[70]
	In vivo studies on male Wistar rats.	↑ Intestinal permeability and lymphatic transport. ↑ AUC (11.52-fold) and C_max._ (19.14-fold) in rat plasma treated with SME1. ↑ AUC (1.95-fold) and C_max._ (5-fold) in rat plasma treated with SME2. The t1/2 and T_max_: SME1= SM2= RSV free. ↑ Oral bioavailability in rat plasma treated with SME1 (76.1%) and SME2 (12.9%).	
**Micelles**			
Bile acids micellar solutions.	In vitro solubility studies in buffer solutions of Na-salts of different bile acids; studies on red blood cells from rabbit	↑ Solubility Micellar solution of 3,7,12-triketocholic acids have lowest membranolytical potential and biggest affinity for RSV solubilization.	[72]
Micellar solution of vineatrol30.	In vitro studies on Caco-2 human intestinal cell line.	↑ Permeability through the cell membrane.	[73]
	In vivo studies on twelve healthy volunteers (six women and six men).	↓ Biotransformation during transit through the enterocytes. ↑ AUC (5-fold) and C_max_ (10.6-fold) vs. vineatrol30, in all subjects. ↓ *T*_max_ in all subjects. ↑ Urinary excretion (4.5-fold) vs. vineatrol30, in all subjects. ↑ Oral bioavailability (5- fold) vs. vineatrol30.	
Casein nanoparticles encapsulating RSV	In vitro release study using simulated gastric (SGF) and intestinal (SIF) fluids; stability studies by measurements of particle size, polydispersity index and zeta potential.	Controlled release rate. ↑ Stability	[74]
	In vivo studies on Male Wistar rats.	↑ Capability to reach the intestinal ephitelium Promotion of RSV intestinal lymphatic transport. ↑ Absorption. ↑ AUC (9.8-fold) and C_max_ (1.45-fold). ↑ t1/2 (9-fold) and T_max_ (3- fold). ↑ Oral bioavailability (10- fold).	
**Polymeric Nanoparticles**			
RSV-poly(lactic-co-glycolic acid) (PLGA) nanoparticles (RSV-PLGA-NPs).	In vitro release study by dialysis bag method; stability studies by measurements of particle size, polydispersity index and zeta potential and encapsulation efficiency; studies on HepG2 human hepatoma cell line.	↑ Solubility and stability. ↑ Encapsulation efficiency and drug loading for RSV. Sustained and slow RSV release from RSV-PLGANPs. ↑ Cellular uptake. ↑RSV-PLGA-NPs bioactivity (lipogenesis reduction, lipolysis promotion and hepatocellular proliferation reduction).	[79]
RSV-loaded galactosylated PLGA nanoparticles (RSV-GNPs)	In vitro release study by dialysis bag method; stability studies by measurements of particle size, polydispersity index, zeta potential and encapsulation efficiency; studies on Caco-2 human intestinal cell line; RAW 264.7 macrophage cell line.	Slower drug release in water. ↑ Stability and entrapment efficiency. ↑ Cellular uptake of RSV-GNPs. ↑ Bioactivity (anti-inflammatory efficacy).	[81]
	In vivo studies on Sprague-Dawley rats.	↑Permeability and intestinal absorption after oral administration. ↑ AUC (2-fold) and C_max_ (1.8- fold). ↑ Oral bioavailability (2-fold).	
**Solid Dispersions**			
Solid dispersion of RSV on Magnesium DiHydroxide (RSV@MDH)	In vitro solubility study by dissolution test in simulated gastric environment.	↑ Solubility and dissolution rate.	[59]
	In vivo studies on New Zealand White hybrid rabbits.	↑ In vivo absorption of RSV from RSV@MDH. ↑ AUC (3.3-fold) and C_max_ (1.33-fold). ↑ T_max_ (2-fold). ↑Oral bioavailability (3-fold).	
GPEDP (grape peel extract-loaded dripping pill) into a solid dispersion	In vitro solubility profile by HPLC; dissolution study by the paddle method.	↑ Solubility and dissolution rate.	[58]
	In vivo studies on Sprague-Dawley rats.	↑ Absorption (12-fold). ↑ AUC (1.92-fold) and C_max_ (7-fold) vs. GPE. ↑ Oral bioavailability (12 fold higher) vs. GPE.	
**Nanocrystals**			
RSV nanocrystals (NCs)	In vitro dissolution study by dialysis bag diffusion method; stability studies by measurements of particle size, polydispersity index, zeta potential and drug content remained; studies on MDA-MB231 breast cancer cell line.	↑ Solubility, stability and dissolution rate. NCs enhanced the RSV delivery in the cells. ↑ RSV bioactivity in NCs form (cell cytotoxicity increase, cell cycle arrest, and apoptosis induction).	[61]
	In vivo studies on Sprague-Dawley rats.	↑ Absorption and uptake across the intestinal barrier. ↑ AUC (3.5-fold) and C_max_ (2.2-fold).	

**Table 3 ijms-20-01381-t003:** Chemical structures of resveratrol derivatives tested in in vitro and/or in vivo models.

Resveratrol Derivatives	References	Resveratrol Derivatives	References
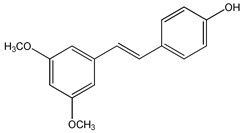 *trans*-3,5-dimethoxy-4′-hydroxystilbene (pterostilbene) **(1)**	[98,99,100,101,102,103,104,105,106,107,108,109,110,111,112,113,114,115,116,117,118,119]	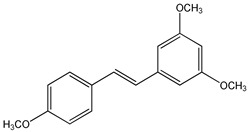 *trans*-3,4′,5-trimethoxystilbene **(2)**	[120,121,122,123,124]
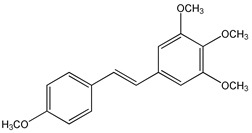 *trans*-3,4,5,4′-tetramethoxystilbene (DMU 212 or TMS) **(3)**	[92,125,126,127,128,129,130,131,132]	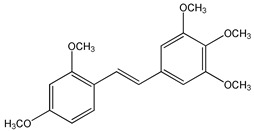 *trans*-2,4,3′,4′,5′-pentamethoxystilbene (PMS) **(4)**	[132,133,134,135,136]
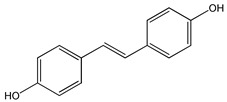 *trans*-4,4′-dihydroxystilbene (DHS) **(5)**	[94,137,138,139,140,141,142,143]	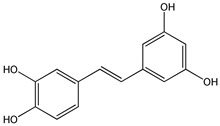 *Trans*-3′,4′,3,5-tetrahydroxy-stilbene (piceatannol) **(6)**	[144,145,146,147,148,149,150,151,152,153,154,155,156,157,158,159,160,161,162,163,164,165]
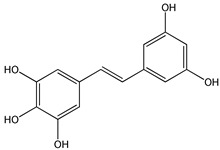 3,3′,4,5,5′-pentahydroxy-trans-stilbene (PHS) **(7)**	[166,167]	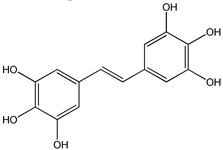 3,3′,4,4′,5,5′-hexahydroxystilbene **(8)**	[168,169,170,171,172]
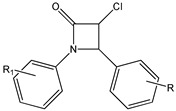 2,3-thiazolidin-4-one RSV derivatives **(9)**	[173,174]	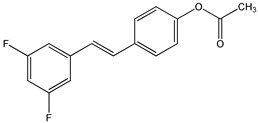 (E)-3,5-difluoro-4′-acetoxystilbene **(10)**	[96]
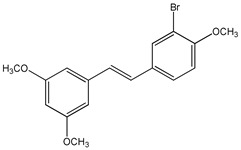 3,4,5-trimethoxy-4′-bromo-*cis*-stilbene **(11)**	[175]	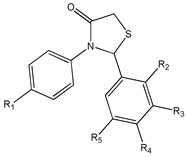 2,3-thiazolidin-4-one RSV derivatives **(12)**	[173,176]
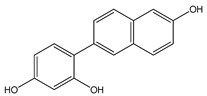 4-(6-hydroxy-2-naphthalen-2-yl)-1,3-benzenediol (HS-1793) **(13)**	[177,178]	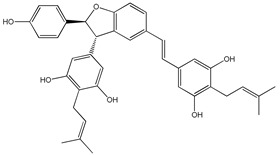 [5-((E)-2-(3-(3,5-dihydroxy-4-(3-methylbut-2-en-1-yl)phenyl)-2-(4-hydroxyphenyl)-2,3-dihydrobenzofuran-5-yl)vinyl)-2-(3-methylbut-2-en-1-yl)benzene-1,3-diol] **(14)**	[179]
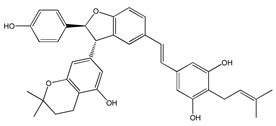 [5-((E)-2-(3-(5-hydroxy-2,2-dimethylchroman-7-yl)-2-(4-hydroxyphenyl)-2,3-dihydrobenzofuran-5-yl)vinyl)-2-(3-methylbut-2-en-1-yl)benzene-1,3-diol] **(15)**	[179]

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
