# Peer review of "Progress to Improve Oral Bioavailability and Beneficial Effects of Resveratrol"

_ijms, 2019, doi:10.3390/ijms20061381_

Reviewer 1 Report

This manuscript is a review article covering the literature on resveratrol  (RSV) bioavailability enhancement. The review is briefly written, covering more than 150 articles. Good thing >50% of the articles covered are published in last 10 years, which is good. The manuscript broadly describes the methodological approaches and synthetic approaches. The focus is more towards synthetic approaches in this manuscript while methodological approaches are briefly covered.     The synthetic approaches cover some derivatives of the RSV.

They any many articles published recently on RSV bioavailability enhancement. This manuscript in present form needs lot of revision for improvising the overall understandability and readability to attract the readership.

 In general, the manuscript English need moderate correction throughout manuscript. The numbers of articles covered in are justified, while the presentation needs thorough revisions. It lacks figures and tables.

 Bioavailability is very broad term. The route of administration and metabolism through respective route can strongly alter the bioavailability. Please specify the route of delivery in title, which this manuscript aimed to cover. If not, then classify/subsection the various routes for which bioavailability enhancement has been reported in literature

 I strongly recommend adding in the tables for the methodological and synthetic approaches.

 Currently, the focus seems to be majorly on synthetic approaches. Therefore, please improve the contents of methodological approaches. Addition of tables and/OR figures in the methodological section will surely help.  In addition, classifying/sub-sectioning the content of methodological approaches can also improve the understandability.

 Conclusion is almost a page. Please avoid discussion, make it concise, and clear.

 Author Response

This manuscript is a review article covering the literature on resveratrol  (RSV) bioavailability enhancement. The review is briefly written, covering more than 150 articles. Good thing >50% of the articles covered are published in last 10 years, which is good. The manuscript broadly describes the methodological approaches and synthetic approaches. The focus is more towards synthetic approaches in this manuscript while methodological approaches are briefly covered.     The synthetic approaches cover some derivatives of the RSV. They any many articles published recently on RSV bioavailability enhancement. This manuscript in present form needs lot of revision for improvising the overall understandability and readability to attract the readership. In general, the manuscript English need moderate correction throughout manuscript. The numbers of articles covered in are justified, while the presentation needs thorough revisions. It lacks figures and tables.

We are grateful for the reviewer comments. In general, the new version of English manuscript has been revised by native speaker and supplemented with tables as suggested. Particularly, we added a table for resveratrol biological effects (Table 1) and tables for methodological approaches (Table 2) and synthetic derivatives (Table 3).

 Bioavailability is very broad term. The route of administration and metabolism through respective route can strongly alter the bioavailability. Please specify the route of delivery in title, which this manuscript aimed to cover. If not, then classify/subsection the various routes for which bioavailability enhancement has been reported in literature.

The manuscript aimed to discuss the resveratrol bioavailability after oral administration. In the new version we specified the route of delivery in the title that is modified as follows: “Progresses in improving oral resveratrol bioavailability and beneficial effects”.

 ·         I strongly recommend adding in the tables for the methodological and synthetic approaches.

As suggested, in the new version, the tables for methodological  approaches (Table 1) and synthetic derivatives (Table 2) have been added.

 ·         Currently, the focus seems to be majorly on synthetic approaches. Therefore, please improve the contents of methodological approaches. Addition of tables and/OR figures in the methodological section will surely help.  In addition, classifying/sub-sectioning the content of methodological approaches can also improve the understandability.

In the new version, in order to improve the understandability of methodological approaches contents, the paragraph 3 “Methodological approaches to improve RSV oral bioavailability” has been re-written and it has been subdivided into sub-paragraph (i.e. 3.1, 3.2 etc.).

 ·         Conclusion is almost a page. Please avoid discussion, make it concise, and clear.

As suggested, in the new version, the conclusions have been rewritten.

 Reviewer 2 Report

Comments:

This manuscript, ID ijms-449130, titled “Progresses in improving resveratrol bioavailability and beneficial effects" for the International Journal of Molecular Sciences. This is a review of the current state of the art of formulations for resveratrol, to improve its bioavailability. During recent years there have been a good number of published reviews focusing on the use of resveratrol for treating a long list of diseases and conditions. This review focuses on strategies of improving the therapeutic bioavailability using nanoformulations and analogs. I think this a much-welcomed focus for a resveratrol review article. The reviewed literature is comprehensive and focused on the topics at hand. My suggestions involve only improvements on the readability and flow of the manuscript.

1.   Page 3, lines 114-116: This paragraph should provide a better transition to the discussion about nanoformulations and analogs.

2.   Page 6, starting in line 214: This is the start of section 2.2, which covers resveratrol analogs. It starts with the discussion about the synthesis of these analogs. This section should start with an introduction, describing the benefits of using analog over the resveratrol itself.

Author Response

This manuscript, ID ijms-449130, titled “Progresses in improving resveratrol bioavailability and beneficial effects" for the International Journal of Molecular Sciences. This is a review of the current state of the art of formulations for resveratrol, to improve its bioavailability. During recent years there have been a good number of published reviews focusing on the use of resveratrol for treating a long list of diseases and conditions. This review focuses on strategies of improving the therapeutic bioavailability using nanoformulations and analogs. I think this a much-welcomed focus for a resveratrol review article. The reviewed literature is comprehensive and focused on the topics at hand. My suggestions involve only improvements on the readability and flow of the manuscript.

We are grateful for the reviewer comments.

·      Page 3, lines 114-116: This paragraph should provide a better transition to the discussion about nanoformulations and analogs.

Following the suggestions of this referee and other referees, we have re-written this part of paper (see new paragraph 2, 3 and 4).

 ·      Page 6, starting in line 214: This is the start of section 2.2, which covers resveratrol analogs. It starts with the discussion about the synthesis of these analogs. This section should start with an introduction, describing the benefits of using analog over the resveratrol itself.

In the new version of paper, we have added the introduction as requested in the paragraph 4

Reviewer 3 Report

The manuscript needs revision for errors in written English. 

Figure 2: Please change the font's reflection in the figure. 

Authors can include a paragraph on the limitations and similarities of the different approaches. 

Discuss/Include pharmacokinetic parameters for different formulations and relative bioavailability compared to native RSV.

Author Response

We are grateful for the reviewer comments.

·         The manuscript needs revision for errors in written English. 

The new version of English manuscript has been revised by native speaker.

 ·         Figure 2: Please change the font's reflection in the figure. 

In the new version, the figure 2 has been changed following this suggestion.

 ·         Authors can include a paragraph on the limitations and similarities of the different approaches. 

In the new version, you can find this aspect in paragraph 3.7

 ·         Discuss/Include pharmacokinetic parameters for different formulations and relative bioavailability compared to native RSV.

In the new version, pharmacokinetic parameters for different formulations and relative bioavailability compared to native RSV are discussed in the paragraph 3 and summarized in the Table 2.

Round  2

Reviewer 1 Report

Accept

Author Response

English language and style of Manuscript are checked. 

Reviewer 3 Report

"In the new version, pharmacokinetic parameters for different formulations and relative bioavailability compared to native RSV are discussed in the paragraph 3 and summarized in the Table 2" This reviewer could not find the relative bioavailability information for different formulations in Table 2. Please clarify. 

Please include common pharmacokinetic parameters for different formulation techniques. 

Some information on the study models and design need to be incorporated. 

Consider changing/restructuring the title of the manuscript.

Author Response

English language and style of manuscript are checked. 

"In the new version, pharmacokinetic parameters for different formulations and relative bioavailability compared to native RSV are discussed in the paragraph 3 and summarized in the Table 2" This reviewer could not find the relative bioavailability information for different formulations in Table 2. Please clarify. 

Following reviewer’s suggestions, table 2 has been modified including the relative bioavailability information for different formulations.

   Please include common pharmacokinetic parameters for different formulation techniques. 

Following reviewer’s suggestions, table 2 has been modified, it now includes the common pharmacokinetic parameters for different formulation techniques.

Some information on the study models and design need to be incorporated. 

Following reviewer’s suggestions, the new table 2 contains more information from  the in vitro and in vivo models. 

 Consider changing/restructuring the title of the manuscript.

Following reviewer’s suggestions, the title has been modified as follows:

“Progress to improve oral bioavailability and beneficial effects of resveratrol”

 Round  3

Reviewer 3 Report

Consider citing some other compounds in the discussion where some of the techniques used for RSV has been previously used (curcumin).  

e.g. https://www.ncbi.nlm.nih.gov/pubmed/30006023